# FUS Alters circRNA Metabolism in Human Motor Neurons Carrying the ALS-Linked P525L Mutation

**DOI:** 10.3390/ijms24043181

**Published:** 2023-02-06

**Authors:** Alessio Colantoni, Davide Capauto, Vincenzo Alfano, Eleonora D’Ambra, Sara D’Uva, Gian Gaetano Tartaglia, Mariangela Morlando

**Affiliations:** 1Department of Biology and Biotechnologies “Charles Darwin”, Sapienza University of Rome, 00185 Rome, Italy; 2Center for Life Nano- & Neuro-Science, Fondazione Istituto Italiano di Tecnologia (IIT), 00161 Rome, Italy; 3Centre for Human Technologies (CHT), Istituto Italiano di Tecnologia (IIT), 16152 Genova, Italy; 4Department of Pharmaceutical Sciences, Department of Excellence 2018–2022, University of Perugia, 06123 Perugia, Italy

**Keywords:** FUS, circRNAs, P525L, ALS, back-splicing, motor neurons, ceRNA, Alu, ADAR

## Abstract

Deregulation of RNA metabolism has emerged as one of the key events leading to the degeneration of motor neurons (MNs) in Amyotrophic Lateral Sclerosis (ALS) disease. Indeed, mutations on RNA-binding proteins (RBPs) or on proteins involved in aspects of RNA metabolism account for the majority of familiar forms of ALS. In particular, the impact of the ALS-linked mutations of the RBP FUS on many aspects of RNA-related processes has been vastly investigated. FUS plays a pivotal role in splicing regulation and its mutations severely alter the exon composition of transcripts coding for proteins involved in neurogenesis, axon guidance, and synaptic activity. In this study, by using in vitro-derived human MNs, we investigate the effect of the P525L FUS mutation on non-canonical splicing events that leads to the formation of circular RNAs (circRNAs). We observed altered levels of circRNAs in FUS^P525L^ MNs and a preferential binding of the mutant protein to introns flanking downregulated circRNAs and containing inverted Alu repeats. For a subset of circRNAs, FUS^P525L^ also impacts their nuclear/cytoplasmic partitioning, confirming its involvement in different processes of RNA metabolism. Finally, we assess the potential of cytoplasmic circRNAs to act as miRNA sponges, with possible implications in ALS pathogenesis.

## 1. Introduction

Fused in sarcoma (FUS) is a DNA/RNA-binding protein playing crucial roles in RNA metabolism and in numerous cellular processes, especially in nerve cells [1,2,3]. Mutations in the FUS gene are indeed linked to the pathogenesis of Amyotrophic Lateral Sclerosis (ALS) [4,5], a neuronal disorder characterized by progressive motor neuron (MN) degeneration that ultimately leads to fatal paralysis [6]. A conspicuous number of the ALS-linked FUS mutations occurs in the C-terminal nuclear localization sequence, resulting in the accumulation of the mutant protein in the cytoplasm [7]. Here, it acquires the ability to form clusters, both in the MN cell body and at synapses [8,9,10], trapping different kinds of RNAs and proteins including pre- and post-synaptic components [10,11,12]. In the cytoplasm, mutant FUS also gains toxic functions [13]—for instance, by binding the 3′UTRs of specific mRNAs encoding for cytoskeletal proteins and other RNA-binding proteins (RBPs)—contributing to the dysregulation of neuronal proteostasis occurring in ALS disease [14,15]. Importantly, the preferential cytoplasmic localization of mutant FUS severely reduces its nuclear functions, including regulation of transcription and processing of coding and non-coding RNAs [1,16,17,18]. In particular, the impact of FUS loss of function on splicing regulation in ALS has been extensively investigated. It has been demonstrated that FUS controls alternative splicing of key neuronal genes, especially of those containing long introns, favoring intron retention, and affects minor intron-containing genes with important function in neurogenesis, dendritic development and action potential transmission in skeletal muscles [1,19,20,21]. More recently, FUS has also been described to control non-canonical splicing events called back-splicing [18]. This latter process joins a downstream donor site with an upstream acceptor site resulting in the formation of covalently closed circular RNAs (circRNAs). To date, beside the fact that the introns involved in back-splicing are usually very long [22], two intronic elements are known to enable circularization: long inverted repeats (mainly Alu repeats [22]) and binding sites for RBPs, both favoring the close proximity of the two splice sites [23]. Notably, in murine in vitro-derived MNs, FUS acts as an RBP affecting the biogenesis of a subset of neuronal circRNAs through the binding to intronic sequences bordering exons that undergo circularization [18].

Circular RNAs (circRNAs) belong to the long non-coding RNA family, and they hold high stability because of their unique closed structure. They are evolutionarily conserved RNAs and are broadly expressed across the animal kingdom [24,25]. Even though they are expressed in every organ, there is a peculiar high accumulation of these transcripts in neuronal tissues [25,26]. Remarkably, several lines of evidence point towards an important role played by circRNAs in nervous system specification and function: they are upregulated during neuronal differentiation and development and are enriched at synapses, where their expression is modulated upon synaptic activity [27,28,29]; most importantly, examples of deregulation of specific circRNAs in neurological disorders, such as Parkinson’s [30] and Alzheimer’s [31,32], have been described [33]. Nevertheless, due to their latest identification, the biological function of circRNAs is still underestimated. Most of them accumulate in the cytoplasm where they can sponge microRNAs [34] and proteins [35], regulating their activity and/or availability, or can be templates for translation [36,37,38]. Nuclear-located circRNAs have also been identified, and their function has been associated with transcriptional control through the interaction with transcriptional machinery or by the formation of an R-loop with the host gene locus [39,40,41].

In this study, we explored the transcriptome of progenitor and mature MNs derived from induced pluripotent stem cells (iPSCs) to assess the effect of the P525L FUS mutation on circRNA biogenesis. This mutation is associated with a strong delocalization of the protein in the cytoplasm and is linked to one of the most severe forms of familial ALS [42,43]. We observed that the highest circRNA production occurs at the stage of neuronal precursors, while a significant decrease was observed in mature MNs. However, a specific set of circRNAs is exclusively expressed in this latter cell population, and most of them resulted in being downregulated in MNs carrying the FUS mutation. In addition, by analyzing public FUS PAR-CLIP data [44], we noticed a preferential binding of mutant FUS to introns of downregulated circRNAs that also contain Alu repeats. Notably, these sequences show high editing level in both WT and mutant conditions, and it is well known that the A-to-I editing mediated by ADAR1 on Alu repeats impairs intron pairing and negatively affects circularization [25,45]. Nevertheless, ADAR1 mRNA levels do not vary in mutant conditions, suggesting an alternative molecular mechanism that causally links the binding of mutant FUS to the intron containing edited Alu repeats and the downregulation of circRNAs. Finally, we found that some downregulated cytoplasmic circRNAs might act as sponges for miRNAs whose targets are involved in axonal trafficking and development, cellular signaling and neurotransmission, strengthening the concept of their potential contribution to ALS pathogenesis.

## 2. Results

### 2.1. Identification of circRNAs Expressed in Human Motor Neurons

To identify circRNAs expressed in human motor neurons (MNs) and determine how their levels are modulated in an ALS background, we reanalyzed Total RNA-seq data (GEO accession number GSE94888) previously published [46] by using the find_circ pipeline [47].

In particular, such data were obtained from cells produced upon the neural induction of iPS cells (day 12), divided into MN progenitors (expressing the HB9::GFP reporter gene, here called GFP+ cells) and other cells (GFP-), and from terminally differentiated MNs (day 12 + 7), carrying wild type (WT) or the P525L mutated form of the FUS gene (FUS^WT^ and FUS^P525L^) [43,46] (Appendix A).

We identified 3857 circRNAs expressed in the various cell populations and genetic backgrounds (average Count Per Million [CPM] ≥ 1 in at least one condition) (Figure 1a and Appendix A); notably, the highest number of circRNAs (2421) was observed in the GFP+ population (638 specifically detected in this cell type). Compared with this population, pure MNs showed a significant drop in circRNA production (1931 and 1833 circRNAs in FUS^WT^ and FUS^P525L^, respectively). Nevertheless, a significant amount of circRNAs (935) was exclusively found in pure MNs, many of them (383) being solely expressed in FUS^WT^ conditions. These observations strongly support the notion of cell- and stage-specific expression of circRNAs [25,48]. When we focused on the host genes, we found that, regardless of the cell type, the vast majority of them are protein-coding (Figure 1b) and most of the circRNAs arise from the coding part of mRNAs (Figure 1c and Appendix A). By comparing them with randomly generated controls, we found that circRNAs preferentially include mRNA regions spanning 5’UTR and coding sequence (CDS) (Appendix A; chi-squared test *p*-value < 1 × 10^−13^ for all the conditions), suggesting a potential coding ability for many of them.

We focused on circRNAs expressed in FUS^WT^ or FUS^P525L^ pure MNs and carried out a differential expression analysis to detect circRNAs deregulated in the cells carrying the mutant protein. We found 121 deregulated circRNAs (*p*-value < 0.05) and, among them, the majority (86) were downregulated in the mutant condition (Figure 2 and Appendix A). We also quantified the linear counterparts in FUS^WT^ and FUS^P525L^ MNs, using the reads spanning linear splicing junctions, and evaluated their differential expression.

The plot in Figure 2a shows that few circRNAs (10 out of 86 downregulated circRNAs and 5 out of 35 upregulated circRNAs) significantly vary in the same direction as their cognate linear RNAs (in order to be less stringent when detecting such concordant circRNAs, we set the *p*-value threshold for linear RNA differential expression to 0.1) (Appendix A). This finding suggests that the alteration of the biogenesis of these circRNAs induced by the FUS mutation occurs preferentially at the post-transcriptional level. 

The Gene Ontology (GO) term enrichment analysis performed using the DAVID tool [49] was conducted on genes hosting downregulated and upregulated circRNAs. The results indicate that downregulated circRNAs arise from genes mainly involved in transcription and apoptosis, while genes hosting upregulated circRNAs are enriched in functions related to chemical synaptic transmission and cytokine-mediated signaling pathways (Figure 2b). Notably, alterations of all these functions are hallmarks of ALS pathology both in cellular models and in clinical settings [50,51,52,53,54,55].

### 2.2. Mutant FUS Binds to Introns Containing Alu Repeats and Is Associated with Downregulated circRNAs

Since FUS has been previously described to act in the biogenesis of specific circRNAs in murine MNs by binding to introns bordering circularizing exons [18], we analyzed public FUS PAR-CLIP data performed on iPSCs-derived FUS^WT^ and FUS^P525L^ MNs [44] to verify whether this mode of action was also conserved in human. Notably, we found that the mutant protein, but not the WT, is preferentially bound to intronic regions flanking circRNAs downregulated in FUS^P525L^ MNs (one sided Fisher’s exact test *p*-value = 0.024, Figure 3a). Moreover, we noticed that such binding is most likely to occur when inverted Alu repeats are hosted in the flanking introns (hypergeometric test *p*-value = 0.038, Figure 3a). Two examples of such co-occurrence of mutant FUS binding sites and inverted Alu repeats are reported in Figure 3b. These data indicate that the mutant FUS protein still residing in the nucleus could negatively affect circRNA biogenesis, somehow altering the Alu-mediated circularization.

Since exon circularization has been shown to be affected by RNA editing [25,45], and Alu elements are often edited by ADAR1 [56,57,58,59], we computed the Alu editing index in FUS^WT^ and FUS^P525L^ MNs—i.e., we calculated the weighted average editing level across all expressed Alu elements in each MN sample. This analysis did not reveal any difference in the global editing level between WT and mutant conditions (Appendix A); furthermore, ADAR1 RNA levels were unchanged (Appendix A). However, when we focused on Alu sequences within 1000 nt intronic regions flanking circRNAs, we found that, when compared with circRNAs not affected by the FUS mutation, downregulated circRNAs have higher editing levels, both in FUS^WT^ and FUS^P525L^ conditions. In contrast, upregulated circRNAs did not show significant changes (Appendix A). These results confirm that RNA editing has an adverse impact on exon circularization, even though we cannot conclude any causal link between the presence of mutant FUS, the increase of the editing levels in these regions and the downregulation of the circRNAs.

### 2.3. Validation of circRNA Expression in MNs

We next focused on seven circRNAs (Table 1) expressed in pure MNs and significantly downregulated in FUS mutant conditions in a discordant manner with respect to the linear counterparts. We performed three independent differentiation experiments taking advantage of an iPSC line (iPSC-NIL) with stable integration of an inducible vector for the expression of Ngn2-F2A-Isl1-T2A-Lhx3 and of a new differentiation protocol [61]. This latter allows the fast conversion of human iPSCs into spinal MNs with high efficiency, avoiding the FACS-sorting step and thus leading to a higher number of cells suitable for all the biochemical investigations conducted in this study.

Using MNs in vitro-derived from FUS^WT^ and FUS^P525L^ iPSCs-NIL (day 12; Figure 4a) we confirmed the deregulation obtained from the RNA-Seq analysis for all the selected circRNAs (Figure 4b). We also observed that both the circRNAs and their linear counterparts were upregulated during MN differentiation (Figure 4c). The only exceptions are circCARHSP1 and circPSME3, which do not show any increase; however, it is interesting to note that their linear counterparts are instead downregulated upon MN differentiation, suggesting an independent biogenesis of the two transcript isoforms (Figure 4c). 

In addition to the circRNAs selected through the RNA-Seq analysis, we decided to study two additional molecules, circSLC8A1 (downregulated in FUS^P525L^ MNs as the linear counterpart) and circDLC1 (downregulated in FUS^P525L^ MNs with a *p*-value just above the significance level), since they hold interesting features. CircSLC8A1 was already described as being deregulated in another neurodegenerative disorder (Parkinson’s disease [30]) possibly contributing to the disease phenotype by sponging miR-128. On the other hand, circDLC1 has been detected in murine MNs [18] and it is one of the most enriched circRNAs in this cell type. Notably, quantitative analysis showed that circDLC1 is significantly downregulated in FUS^P525L^ MNs, while circSLC8A1 has a *p*-value very close to significance (Figure 4b); however, both circRNAs were upregulated during MN differentiation (Figure 4c). Moreover, circDLC1 is one of the most enriched circRNAs also in human MNs (Appendix A).

In order to exclude that circRNA downregulation was due to differences in MNs maturation or FUS (WT and P525L) expression between the two conditions, we also measured the levels of CHAT, ISLET1, and FUS mRNAs in WT and mutant MNs. Similar levels of these mRNAs were observed in both conditions, thus confirming that circRNA deregulation is a real consequence of the presence of ALS-linked FUS mutation (Appendix A).

When assessed for circularity, all the putative circRNAs showed resistance to the RNase R treatment, albeit to different extent, while the linear counterparts were almost completely degraded (Figure 5a). circSLC8A1 and circSMARCA5 were excluded from this analysis, since their circularity has already been verified in previous studies [30,41].

We next investigated whether the variation of the levels of the studied circRNAs was specifically caused by the P525L mutation or if it was a more general feature linked to the ALS condition. To this aim, we analyzed two additional RNA sequencing datasets (NCBI GEO: GSE203173 and dbGaP: phs000747) obtained from in vitro derived MNs carrying the H517Q FUS mutation and from sporadic ALS post-mortem tissues (ventral horns of the lumbar spinal cords), respectively [62,63]. As shown in Appendix A, a significant alteration of circRNA levels was observed for both the FUS^H517Q^ (76 downregulated and 226 upregulated circRNAs) and ALS samples (363 downregulated and 271 upregulated circRNAs); however, a very poor overlap was observed between the circRNAs deregulated in these datasets and in the FUSP^525L^ MNs (Appendix A). Among the circRNAs selected in the present study, only circCORO1C and circCARHSP1 were found to be altered in the FUS^H517Q^ condition, while no overlap was observed with the ALS tissues. Nevertheless, they showed an opposite trend of deregulation, as observed for the vast majority of circRNAs deregulated in both FUS mutants. Moreover, it is worth noting that, while in FUS^P525L^ MNs only 12% of the deregulated circRNAs vary in a concordant manner with respect to their linear counterparts, in FUS^H517Q^ MNs and in ALS tissues a higher level of concordance was observed (29% and 33%, respectively), suggesting a major contribution of transcriptional regulation in these latter conditions. Notably, transcriptional defects have been already observed in MNs carrying the H517Q FUS mutation [62].

These results lead to conclude that the P525L mutation causes a specific deregulation of a subset of circRNAs not observed in other ALS conditions and strengthen the indication that the P525L-linked deregulation mainly occurs at the post-transcriptional level.

### 2.4. CircRNAs Have Preferential Cytoplasmic Localization

To get insight into the function of the selected circRNAs, we explored their subcellular localization by carrying out biochemical nuclear/cytoplasmic fractionation of WT and mutant MNs. The levels of the mature and precursor GAPDH transcripts in the two separated cellular compartments indicate the high efficiency of the procedure (Figure 5b). We observed that the majority of circRNAs are significantly enriched in the cytoplasm (CYT versus NUC *p*-value < 0.05, two-tailed Student’s *t*-test), except for circSMARCA5 and circZNF124, which resulted in being located predominantly in the nucleus (Figure 5b). Notably, while circSMARCA5 has been already described as being localized and to act in the nucleus [41], no evidence exists regarding the localization and function of circZNF124 in this compartment.

Since the P525L FUS mutation causes a prominent delocalization of the protein in the cytoplasm of MNs, we tested whether, besides affecting circRNA expression levels, mutant FUS might also have an impact on their nuclear/cytoplasmic partitioning [43].

Notably, the localization of three circRNAs, circCARHSP1, circNAA35, and circPSME3, significantly shifted towards the cytoplasm in FUS^P525L^ MNs. This could have interesting implications for circRNA function, as already shown for circHdgfrp3 in murine MNs [11]. We also noticed a shift in the localization for the nuclear circSMARCA5 and circZNF124, even though not significant. However, it is worth noting that, in WT conditions, circSMARCA5 and circZNF124 are significantly enriched in the nucleus, while in FUS mutant conditions this enrichment lacks significance (Appendix A), suggesting that also in this case there might be a shift in the localization promoted by the FUS mutation that impinges on the downregulation observed in the same pathological condition.

### 2.5. Prediction of a circRNA–miRNA–mRNA Network Altered in FUS^P525L^ MNs

The availability of total and small RNA sequencing data from FUS^WT^ and FUS^P525L^ MNs, as well as proteome data [14,46], allowed us to predict whether downregulated cytoplasmic circRNAs could bind to miRNAs expressed in MNs, thus influencing the expression of their targets, and to infer a circRNA–miRNA–mRNA circuitry (Appendix A).

In this analysis, we considered only those targets whose RNA (from RNA-Seq [46]) and/or protein levels (from mass spectrometry [14]) decrease upon the FUS mutation concordantly with an increased number of free miRNA molecules due to the downregulation of the sponging circRNA. We decided to select cytoplasmic circRNAs having at least two binding sites for a miRNA, or at least one site spanning the back-splicing junction.

Figure 6 shows the predicted regulatory network, which involves 4 circRNAs potentially acting as sponges for the indicated miRNAs. Notably, biological processes that might be regulated by the sponge activity of circRNAs are particularly relevant for ALS pathology. For instance, circSLC8A1 shows four binding sites for miR-335-3p and three for miR-27a-3p, miR-27b-3p and miR-30d-3p, and the mRNA targets of these miRNAs are involved in processes that are altered in ALS such as: neuroprotection (e.g., ADAM12 [64]), DNA damage repair (e.g., HLTF [65,66]), protein trafficking regulation and axonal guidance/elongation (e.g., SNX18 and DPYSL3 [67,68]), splicing regulation (e.g., EXOC7 and FUBP1 [69,70]), neuronal excitability and synaptic transmission (e.g., NEDD4L [71]). Remarkably, binding sites for miR-335-3p have been found both in circSLC8A1 and in circDLC1, and all circRNAs, by sponging different miRNAs, act on genes that have a common molecular function or that are involved in the same processes. Altogether these observations suggest a cooperative mode of action with the aim to reinforce the regulatory potential of a single circRNA–miRNA–mRNA axis.

CircDlc1 also binds to miR-217 which, among the others, targets HNRNPA3. Notably, reduced levels of HNRNPA3 protein in C9orf72 ALS cases have been associated with enhanced repeat-dependent toxicity [72] and, more recently, HNRNPA3 has also been described as a modifier of FUS toxicity [73]. Moreover, mutations in the HNRNPA3 gene have been identified in rare cases of ALS [74].

Another intriguing aspect of circRNA sponge activity is that, upon circularization, new binding sites for miRNAs might be generated at the back-splicing junction. Interestingly, we found that this specific region of circCARHSP1 is targeted by miR-24-3p and miR-345-5p. Notably, miR-24-3p regulates neuronal differentiation and promotes neurite growth [75,76], while miR-345-5p controls viability of microglia cells and represents a promising biomarker of the progression of C9orf72-associated disease [77,78].

Finally, we decided to validate the predicted circRNA sponge activity by downregulating circDLC1 in MNs through siRNA treatment and by using quantitative Real-Time PCR to evaluate the expression of its putative targets as preliminary screening. Among the predicted targets listed in Appendix A we chose ADAM12 and EXOC7 targeted by miR-335-3p and HNRNPA3, ENAH and PDHA1 targeted by miR-217. We observed a reduction only for ENAH mRNA (Figure 6b) pointing to possible regulatory circuitry involving circ-DLC1/mir-217 axis. For the other mRNA targets that do not vary in circDLC1 knock down condition a further analysis of protein levels will be performed in the near future. Moreover, regarding the miR-335-3p targeted mRNAs a cooperative mode of action of circDLC1 and circSLC8A1 has to be considered; therefore, the absence of only one of the two circRNAs might be not sufficient to determine alteration of the target expression.

## 3. Discussion

CircRNAs are rediscovered molecules that are widely expressed and conserved in the animal kingdom [24,25]. They hold a unique structure that confers them high stability and the potential to be used as circulating biomarkers for diseases [33,79]. One of the most interesting features of circRNAs is their remarkable expression in nervous tissues [25,26]. Many studies revealed that they have an extremely specific expression in the different brain regions and their levels are modulated during neurogenesis and neuronal activity [27,28,29]. Notably, levels of some specific circRNAs have been found to be altered in Parkinson’s and Alzheimer’s diseases [30,32] and mice in which the circRNA CDR1as has been knocked out show neurological defects [80]. To date, few studies concerning ALS have shown a correlation between circRNA deregulation and this disease. More recent work showed differential expression of circRNAs in leukocytes from ALS patients [81] and two papers from our laboratory demonstrated the direct involvement of the FUS protein in the biogenesis and localization of a subset of circRNAs expressed in in vitro-derived murine MNs [11,18]. FUS plays an important role in splicing regulation in neurons. Most of the FUS targets are pivotal genes for neuronal maintenance and survival and, more interestingly, among the targets there are genes encoding for RBPs involved in splicing regulation [1,2]. It is therefore not surprising that mutations in the FUS gene cause a profound alteration of the transcriptome and proteome leading to neuronal defects and degeneration. The present study provides additional evidence to this alteration by revealing the misregulation of circRNAs expression in MNs carrying the P525L FUS mutation and by linking their deregulation to ALS pathogenesis. More specifically, we conducted a bioinformatic analysis to identify back-splicing events in RNA-Seq data collected along the stepwise in vitro differentiation of human iPSCs towards the MN fate [46]. This protocol has been performed in WT conditions and in the presence of the FUS P525L mutation. The insertion of GFP reporter gene controlled by the MN specific HB9 promoter in all iPSC lines allowed the discrimination and the purification of MN precursors (GFP+) from the other cell populations (GFP-). Then, pure population of GFP+ precursors has been further maturated (Appendix A). Notably, GFP+ cells show the highest number of expressed circRNAs, consistently with the observation of an upregulation of this type of RNA during neuronal differentiation [25]. Although we observed a decrease in the number of circRNAs in mature MNs, it is noteworthy that the three different cell populations, GFP+, GFP- and mature MNs, display restricted expression of a subset of specific circRNAs, confirming the cell type specificity that characterizes these molecules [25].

By looking at circRNAs deregulated in the mutant MN samples, we have enlarged the repertoire of FUS targets in human neuronal cells, possibly contributing to the unraveling of molecular mechanisms pertaining ALS that are still not completely understood. In particular, we discovered that the FUS mutation leads to the deregulation of 121 circRNAs in MN cells. Notably, in most cases their linear counterparts do not show the same trend, suggesting that mutant FUS is acting at the post-transcriptional level.

Since circRNAs may control the expression of their host genes, GO term enrichment analysis was carried out on genes producing deregulated circRNAs. We found interesting functional categories such as transcription, apoptosis, cell adhesion, and chemical synaptic transmission, which are all described as being altered in ALS [13,46,51,82].

Considering the described ability of FUS to affect circularization by binding intronic sequences [18], we analyzed public FUS PAR-CLIP data obtained from in vitro-derived FUS^WT^ and FUS^P525L^ MNs [44] to examine the binding of FUS to the introns of circRNAs deregulated in FUS^P525L^ MNs. Significant preferential binding of mutant FUS was observed for introns of downregulated circRNAs, usually coinciding with the presence of inverted Alu repeats. The A-to-I editing enzyme ADAR1 has been described to preferentially act on such repeats [56,57,58,59] and altered ADAR1 editing has been linked with neurological disorders [83]. Indeed, although inverted Alu elements have the intrinsic ability to promote circularization, their ADAR1-mediated editing, by inhibiting intron pairing, can reduce back-splicing events [25,45]. The absence of any difference in the expression and activity of ADAR1 between WT and mutant MNs suggests that the alteration in circRNA production is not caused by the direct action of FUS on ADAR1.

Nevertheless, among the circRNAs affected by FUS mutation, only the downregulated ones showed high Alu editing levels in their flanking intronic regions compared to unaffected circRNAs. Further analyses are indeed required to eventually clarify the molecular mechanisms behind the concomitant presence of the P525L FUS protein and edited Alu repeats in the introns of downregulated circRNAs. At this stage, we might just speculate that the presence of mutant FUS on introns enriched for edited Alu repeats allows the recruitment of additional factors that eventually exacerbate the inhibitory effect of editing on circularization.

The study of the subcellular localization of a subset of downregulated circRNAs showed that only two of them have a clear nuclear localization, while all the others are preferentially located in the cytoplasm. P525L mutation causes a profound cytoplasmic delocalization of FUS with consequences on the expression of many genes through a gain of function mechanism [15,84,85,86]. In addition, we recently provided another example of this mode of action by demonstrating that in stress conditions the presence of mutant FUS causes the entrapment of circHdgfrp3 in the body of MNs, reducing its ability to travel along neurites [11]. In the present study we show that also in human MNs, mutant FUS can alter the cytoplasmic-nuclear distribution of selected circRNA molecules, thus possibly altering their function.

The analysis of two additional RNA sequencing datasets obtained from different types of ALS samples (FUS^H517Q^ in vitro derived MNs and ALS post-mortem tissues) allows to conclude that the biogenesis of the circRNAs characterized in this study is specifically affected by P525L FUS mutation, since they do not vary or do not show the same trend of variation in the other two conditions. Moreover, differently from what is observed in FUS^P525L^ MNs, a higher percentage of deregulation likely occurs at the transcriptional levels both in FUS^H517Q^ MNs and ALS tissues.

The discordant results obtained with the two FUS mutations both involving the Nuclear Localization Signal of the protein might be explained by the intrinsic highly specific phenotype observed in ALS patients carrying FUS mutations [87]. More specifically, the P525L mutation is linked to one of the most severe and juvenile form of ALS and leads to an important delocalization of FUS in the cytoplasm, while the H517Q recessive mutation is associated to a milder disease course, with an adult onset, and to a moderate mislocalization phenotype [88]. Therefore, the two FUS mutations probably lead to MN dysfunction through different pathogenic mechanisms altering specific sets of genes with distinct modes of action.

Concerning the functional implication of the deregulation of circRNAs in mutant MNs, we decided to investigate the potential of downregulated cytoplasmic circRNAs to function as miRNA sponges, since many molecular mechanisms associated to ALS involve the action of non-coding RNAs, including miRNAs [89,90]. By integrating RNA sequencing and proteomics data, we built a competing endogenous RNA (ceRNA) network describing how the downregulation of four circRNAs could result in the miRNA-mediated repression of genes whose function have important implications in the pathology of ALS.

circSLC8A1 shows the highest number of miRNA-binding sites, and even though this was not surprising since it is the longest circRNA analyzed in this study, it is noteworthy that it contains multiple binding sites for each miRNA identified. For instance, it has four binding sites for miR-335-3p and three for miR-27a-3p, miR-27b-3p, and miR-30d-3p. More remarkably, the mRNA targets for these miRNAs are involved in processes that are altered in ALS such as: neuroprotection, DNA damage repair, protein trafficking regulation and axonal guidance/elongation, splicing regulation, neuronal excitability and synaptic transmission. Notably, two binding sites for miR-335-3p are also present on circDLC1, suggesting a possible cooperation between the two circRNAs. The lack of variation of miR-335-3p mRNA targets upon circDLC1 downregulation in MNs would indeed indicate the presence of this active cooperation. Therefore, the alteration of the expression of miR-335-3p mRNA targets might be achieved only when both circRNAs are downregulated.

However, circDLC1 has two binding sites also for mir-217 and downregulation of this circRNA in MNs affects the mRNA levels of ENAH, a putative target of this miRNA. This result indicates that the absence of circDLC1 enables mir-217 to downregulate the expression of its targets, confirming the sponge activity of this circRNA versus miR-217. Nevertheless, some other mir-217 targets do not show significant variation indicating that a deeper investigation of protein levels is required. ENAH belongs to the Ena/VASP protein family, is highly abundant in the nervous system and it is required for normal development during neuritogenesis, axon guidance response and synapse formation [91,92,93]. Notably, ENAH controls the translation of many mRNAs and in particular of Dirk1a [94]. This encodes for a kinase playing many roles in neuronal development and was described as being involved in the pathology and etiology of diverse neurological disorders, including Alzheimer’s and Parkinson’s disease [95,96]. These findings strongly support the notion that regulatory circuitries involving circRNAs might have important implications in the neuronal system in physiology and pathology.

Finally, the peculiar biogenesis of circRNAs generates back-splicing junctions (BSJ) that may serve as circular-specific binding sites for miRNAs. Such mode of interaction was indeed detected for circCARHSP; miRNAs bound to its BSJ, miR-345-5p and miR-24-3p, regulate the expression of many genes involved in the endo-lysosomal pathway and autophagy (VAMP2, RAB34, VSP28 and MVB12B [97,98,99]). Notably, alteration of these processes leads to impaired proteostasis, a hallmark of ALS [100].

ALS is a complex and multifactorial disease involving different pathological processes; nowadays, we are well aware that the comprehension of the molecular mechanisms which lead to the alteration of such processes may represent an actual opportunity to foresee clinical treatments. We believe that the present study, even if preliminary, contributes to add insights to this complex scenario by including other players in the regulatory networks involved in the disease and by assessing how the coordinate action of different ncRNAs (circRNAs and miRNAs) may affect the molecular pathogenesis of ALS.

## 4. Materials and Methods

### 4.1. Computational Identification of Back-Splicing Events

RNA-Seq reads were initially trimmed using the Trimmomatic software [101] to remove adapter sequences and poor quality bases; the minimum read length after trimming was set to 18. After that, Bowtie 2 [102] was used to align reads to a sequence database composed of rRNA, tRNA, snRNA, snoRNA, and other non-coding species that resulted in being overrepresented according to the FastQC software [103]. Reads mapping linearly to these sequences were filtered out. The remaining reads were used for circRNA detection as follows: the two mates of each pair of reads (R1 and R2) were aligned separately to the reference genome (Bowtie 2 to GRCh38), and those mapping were discarded; the rest was used as input for find_circ [47]. First, 20 nt long anchors were produced from the ends of each read; then, anchors were aligned to the reference genome with Bowtie 2 and alignments of both R1- and R2-derived anchors were used by find_circ to identify and count circular and linear splicing events, restricted by the presence of a GU/AG signal. Circular splicing events from each biological sample were then filtered based on the following conditions: at least two unique reads mapping to head-to-tail junctions, distance between mapped anchors < 100 kb, and mapping quality of the anchors ≥ 35. Filtered circRNAs from each sample were then merged and for all of them, reads mapping linearly to each of the two coordinates of the head-to-tail splice junction were parsed from the find_circ output and counted. The BEDTools software suite [104] was employed to annotate circRNAs by intersecting their genomic coordinates with those of the genomic features described in the Ensembl 77 human gene annotation [105]. We further excluded those putative back-splicing events that were spanning two non-overlapping genes, which were likely due to mapping errors.

The Venn diagrams showing the localization of circRNAs with respect to gene features were generated using the jvenn tool [106]. The comparison of the observed and the expected localization of circRNAs within the body of protein-coding transcripts was done as follows: circRNAs whose back-splicing junctions fell in an intron or outside the gene were filtered out; we then computed the distribution of the number of exons included between each circRNA back-splicing junction pair (for those cases in which the exonic structure of circRNAs could not be defined unambiguously due to the alternative splicing of the gene, we used the average number of exons rounded to the nearest integer); from the transcripts hosting these circRNAs, we randomly picked 5000 groups of consecutive internal exons (used to simulate faux circRNAs), the number of exons of each group being sampled from the distribution of the number of exons previously computed; the localization of real and faux circRNAs with respect to untranslated regions and coding regions was determined and visualized using the Venny 2.1 web application [107]; a chi-squared test was performed to determine whether real circRNAs show a preferential localization when compared to random faux circRNAs.

### 4.2. CircRNA Quantification and Differential Expression Analysis

To quantify the expression of circRNA species and linear counterparts, we generated a matrix containing the read counts of both the back-splicing events and their cognate linear splicing events; the read counts of cognate linear splicing events were calculated summing all the reads mapping linearly on both the splice junctions involved in back-splicing. Events not having two or more reads in at least two samples were filtered out. The *cpm* function from the edgeR R package [108] was employed to calculate CPM values. circRNAs were considered to be expressed in a condition if the average CPM in that condition was ≥1. The UpSetR R package [109] was used to compute and visualize the intersection of the expressed circRNA sets from the four cell types.

To identify those circRNAs that were differentially expressed between two conditions motor neurons, circRNAs and linear splicing events were initially filtered by removing those not supported by at least 2 reads in a number of samples equal to the minimum number of replicates. edgeR R package was employed to perform differential expression analysis on the resulting count matrix; an additive model was fitted to adjust for baseline differences between the three independent differentiation experiments: *design <- model.matrix (~ experiment + condition)*; model fitting and testing was performed using *glmFIT* and *glmLRT* functions. Given the low number of reads used for testing, we decided to use the *p*-value instead of false discovery rate to select for differentially expressed events, setting the significance threshold value to 0.05 for back-splicing events and to 0.1 for linear splicing events; we chose to use a more relaxed cutoff for linear splicing events to obtain a set of circRNAs whose deregulation was less likely to be due to the concordant change in expression of the host gene.

The DAVID web server [110] was used to evaluate the GO Biological Process terms enriched among the protein-coding host genes of downregulated and upregulated circRNAs, using the list of all the protein-coding host genes of the circRNAs tested for differential expression as a background. Categories with a *p*-value < 0.05 were considered as significantly enriched.

### 4.3. Analysis of circRNA Flanking Intronic Regions

To gain insight into the intronic determinants regulating the biogenesis of the circRNAs whose expression is altered upon the FUS mutation, we focused on the 1 kb-long regions upstream and downstream the circRNAs tested for differential expression; among the deregulated circRNAs, we analyzed only those whose linear counterpart does not vary significantly in the same direction. To assess FUS binding in the intronic regions, we used the publicly available FUS^WT^ and FUS^P525L^ PAR-CLIP data produced from in vitro-derived human MNs [44]. Flanking intronic regions containing at least one T to C transition were marked as bound by FUS; the overlap between intronic regions and PAR-CLIP transition events was assessed using the BEDTools intersect tool [105]. To evaluate whether FUS binding events were enriched in the intronic regions flanking deregulated circular RNAs, we performed one-sided Fisher’s exact tests comparing the proportions of upregulated and downregulated circular RNAs with at least one binding site in the intronic regions to the proportion of non-deregulated circRNAs with such binding.

To evaluate the presence of inverted Alu repeats in the circRNA intronic regions, we first extracted the coordinates of Alu elements from the UCSC genome browser [111] hg38 RepeatMasker [112] annotation. BEDTools intersect was employed to identify the circRNAs carrying at least two Alu elements with opposite orientation, one in the upstream intronic region and the other in the downstream intronic region. The significance of the overlap between the circRNAs with FUS binding sites in the intronic regions and the circRNAs with inverted Alu repeats was assessed by performing hypergeometric tests.

To calculated the Alu editing index, preprocessed FASTQ files produced from Total RNA-Seq analysis of MN FUS^WT^ and FUS^P525L^ samples [46] were given as input to the PRINSEQ software [113] in order to remove exact duplicates and reverse complement exact duplicates; reads where then aligned to the hg38 genome using STAR [114] with parameters *--outSAMtype BAM SortedByCoordinate --outFilterType BySJout --outFilterMultimapNmax 1 --alignSJoverhangMin 8 --alignSJDBoverhangMin 1 --outFilterMismatchNmax 999 --outFilterMismatchNoverLmax 0.04;* finally, the RNAEditingIndexer tool [115] was employed, providing it with the alignment files and the previously computed gene-level FPKM values [46] also used to evaluate the expression of ADAR enzymes). RNAEditingIndexer was also employed to calculate the editing index only for the Alus localized within the circRNA flanking intronic regions. Paired Student’s *t*-tests were conducted to evaluate the Alu editing index difference between deregulated and non-deregulated circRNAs.

### 4.4. Inference of the circRNA–miRNA–mRNA Regulatory Network

In order to infer the full sequences of circular RNAs for miRNA-binding prediction, we first intersected the coordinates of their back-splicing junctions with those of the exons belonging to transcripts expressed in MN samples [46] using BEDTools intersect. For each transcript overlapping with a circRNA, we extracted the subsequence spanned by the circRNA and extended its 3′ end with the first 125 nucleotides from their 5′ end, thus obtaining a database of possible circRNA sequences. We then aligned each pair of back-splicing junction-mapping reads to this sequence database using Bowtie 2 with the *–very-sensitive* option. For each circular RNA, we selected those read pairs that were aligned to the same extended subsequence, with at least one of them mapping across the original 3′ end. The final circular RNA sequence was then assigned by choosing the transcript subsequence with the highest number of mapping read pairs, retaining only the first 25 nucleotides from the extension. Such extended circular RNA sequences were used for miRNA-binding site prediction, performed using the miRanda tool [116] with default parameters; this analysis was restricted to the miRNAs that are expressed in the MN samples [46]. circRNAs with at least two sites for a given miRNA, or with at least one site spanning the back-splicing junction, were considered as putative miRNA sponges. To identify potential targets for sponged miRNAs, we used the miRWalk version 3 [117] web resource, which contains miRNA–mRNA interactions predicted using the TarPmiR algorithm [118] and integrates TargetScan Release 8.0 [119] and miRDB Release 7.0 [120], which are two online databases of predicted miRNA targets, along with miRTarBase Release 8.0 [121], an online database of experimentally supported miRNA-gene interactions; we also employed validated miRNA targets from TarBase version 8 [122]. miRNA-3′ UTR interactions available in miRWalk were selected using a TarPmiR score threshold equal to 0.95 and by requiring the presence of the miRNA-gene pair in at least one of the other four databases mentioned above. The targets of the miRNAs sponged by the selected downregulated cytoplasmic circular RNAs were further filtered to include only those that, upon FUS mutation, are downregulated at the mRNA [46] and/or at the protein [14] level. The function of such targets was explored by performing a GO Biological Process and Molecular Function term enrichment analysis using the WebGestalt 2019 tool [123], providing the list of protein-coding genes that are expressed in MNs and available in miRWalk as a reference set; enriched categories were selected using a *p*-value cutoff equal to 0.01. The results of all these analyses were integrated to construct a circRNA–miRNA–mRNA network, in which miRNAs are connected to putative circRNA sponges, to known/predicted downregulated targets and to the enriched categories to which at least one of their targets belongs. Cytoscape version 3.9.1 [124] was used to visualize this network.

### 4.5. iPSCs Maintenance, Spinal MN Differentiation and Treatment

iPSCs, stably transfected with epB-NIL, an inducible expression vector containing the Ngn2, Isl1, and Lhx3 transgenes [125], were maintained in Nutristem XF/FF medium (Biological Industries, Beit-Haemek, Israel) in plates coated with hESC-qualified Matrigel (BD Biosciences, Franklin Lakes, NJ, USA) and passaged every 4–5 days with 1 mg/mL dispase.

To obtain spinal MNs, iPSCs were differentiated as previously described [61]. Briefly, hiPCSs were dissociated to single cells with Accutase (Thermo Fisher Scientific, Waltham, MA, USA) and plated in Nutristem-XF/FF medium (Biological Industries) supplemented with 10 μM rock inhibitor (Enzo Life Sciences, Pero (MI), Italy) on Matrigel coated dishes (BD Biosciences) at a density of 100,000 cells/cm^2^. After 2 days of 1 μg/mL doxycycline (Thermo Fisher Scientific) induction in DMEM/F12 medium (Sigma-Aldrich, Milan, Italy), supplemented with 1× Glutamax (Thermo Fisher Scientific), 1× NEAA (Thermo Fisher Scientific), and 0.5× Penicillin/Streptomycin (Sigma-Aldrich), the doxycycline induction is maintained for additional 3 days in Neurobasal/B27 medium (Neurobasal Medium, supplemented with 1× B27, 1× Glutamax, 1× NEAA, all from Thermo Fisher Scientific, Waltham, MA, USA; and 0.5× Penicillin/Streptomycin, Sigma Aldrich, Milan, Italy), containing 5 μM DAPT and 4 μM SU5402 (both from Sigma-Aldrich, Milan, Italy). At day 5, cells were dissociated with Accutase and plated on Matrigel-coated dishes at a density of 100,000 cells/cm^2^. Ten micrometers of rock inhibitor were added for the first 24 h after dissociation. Neuronal cultures were maintained in neuronal medium (Neurobasal/B27 medium supplemented with 20 ng/mL BDNF, 10 ng/mL GDNF, both from PreproTech, London, UK, and 200 ng/mL l-ascorbic acid, Sigma Aldrich, Milan, Italy) for 7 days.

Downregulation of circDLC1 was achieved by siRNA transfection at day 5 of differentiation and cells were collected after 3 days. A mix of two siRNAs targeting the back splicing junction of circDLC1 and the control siRNA scramble (see Appendix A) were used at 50 nM final concentration in presence of RNAiMax transfection reagent (Thermo Fisher, Waltham, MA, USA) following the manufacturer’s instructions.

### 4.6. Nucleus/Cytoplasm Fractionation

Nucleus/cytoplasm fractionation was performed on an enriched MN population (day 5 of the iPSCs-NIL differentiation). After treating the cells with accutase for 10 min at 37 °C, the reaction was stopped using DMEM-F12 and the cells were centrifuged at 1200 rpm for 5 min. The pellet was washed twice with DPBS, and the solution was centrifuged for 5 min at 1200 rpm. The supernatant was completely removed, and the cellular pellet was treated with the PARIS kit (Thermo Fisher Scientific, Waltham, MA, USA), according to the manufacturer’s instructions.

### 4.7. RNA Extraction and Analyses

Total RNA was extracted using the Direct-zol Miniprep RNA Purification Kit (Zymo Research, CA, USA) and reverse-transcribed using the SuperScript™ VILO™ cDNA Synthesis Kit (Invitrogen, Thermo Fisher, Waltham, MA, USA), all according to the manufacturer’s instructions. RNaseR treatment was performed on 1 μg of total RNA extracted from MNs at day 12 of differentiation; 2U of RNaseR (RNR07250, Epicentre, Wisconsin, USA) was used and the reaction was carried out for 15 min at 37 °C; the RNA was then extracted and reverse-transcribed as described above. One picogram of a DNA spike-in molecule was added to each reaction for quantitative real-time PCR (qPCR) normalization. DNA spike-in was produced from the multiple cloning site in pcDNA3.1(-) (Thermo Fisher Scientific, Waltham, MA, USA). qPCR was performed by using PowerUp SYBR Green Master Mix (Thermo Fisher Scientific) following the manufacturer’s suggested protocol. Target gene expression was obtained by normalizing target quantity per housekeeping gene ATP5O and GAPDH quantity used as a control gene and the Relative RNA quantity was calculated as the fold change, 2^−ΔΔCt^, or as 2^−ΔCt^. DNA amplification was monitored with an ABI 7500 Fast qPCR instrument. Data analysis was performed using the SDS Applied Biosystem 7500 Fast Real-Time PCR system software. Oligonucleotides are listed in Appendix A.

## Figures and Tables

**Figure 1 ijms-24-03181-f001:**
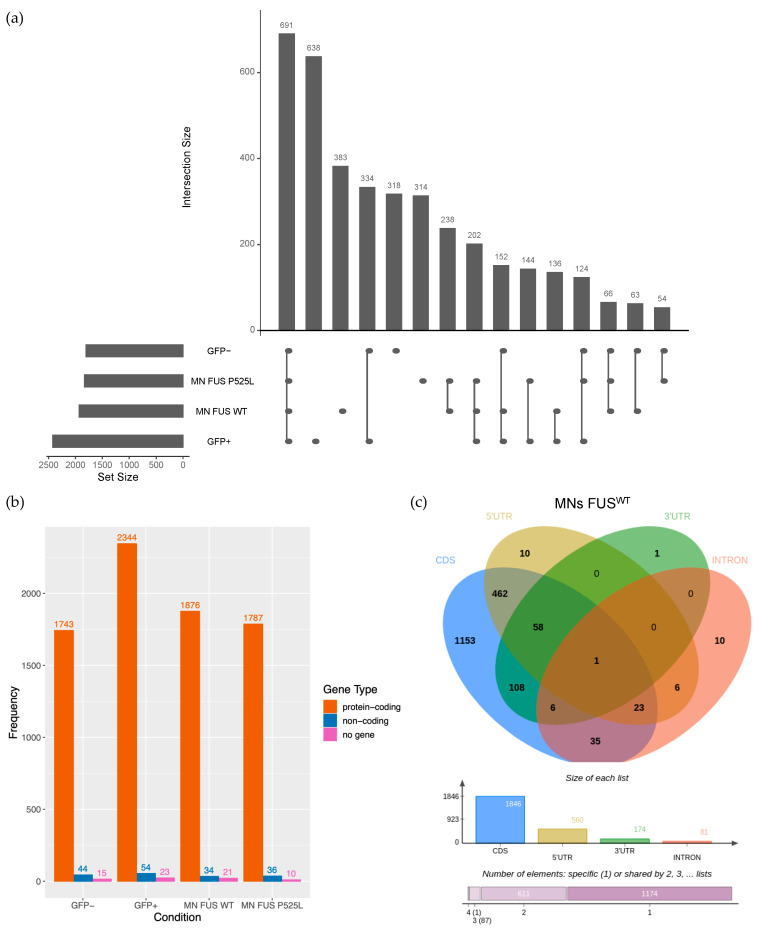
Distribution of circRNAs in iPSCs-derived MN precursors and mature MNs. (**a**) UpSet plot showing the number of circRNAs identified (average CPM > 1) in GFP-, GFP+, FUS^WT^, and FUS^P525L^ cell types, as well as the overlap between such conditions. (**b**) Bar chart showing, for each cell type, the number of circRNAs hosted by protein-coding genes, non-coding genes, and intergenic regions (no gene). (**c**) Venn diagram and bar charts showing the protein-coding gene regions occupied by the circRNAs detected in MN FUS^WT^ samples. CDS: coding sequence; 5′UTR: 5′ untranslated region; 3′UTR: 3′ untranslated region; INTRON: intronic region.

**Figure 2 ijms-24-03181-f002:**
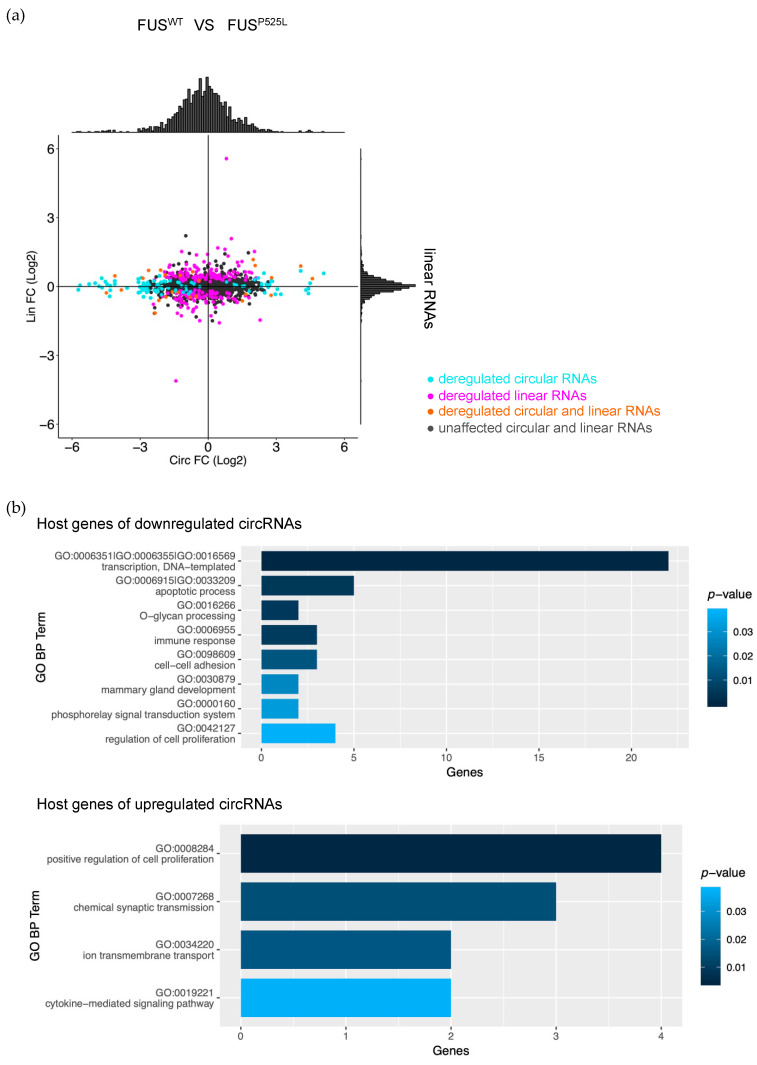
Effect of FUS mutation on circRNA production in mature motor neurons. (**a**) Scatter plot showing the change in the expression of circRNAs (x-axis) and their cognate linear RNAs (y-axis) in FUS^P525L^ MNs with respect to FUS^WT^ MNs. The distributions of the log2 fold change (FC) values are shown above and aside the scatter plot. Turquoise dots are used to indicate when only the circRNA is deregulated (*p*-value < 0.05); magenta dots when only the linear RNA is deregulated (*p*-value < 0.1); orange or black dots when both the circRNA and the linear counterpart are deregulated or unaffected, respectively. (**b**) Enriched GO Biological Process terms identified in the host genes of downregulated and upregulated circRNAs. The x-axis represents the number of host genes belonging to the category reported in the y-axis. Color intensity is proportional to the enrichment *p*-value. Redundant categories were manually clustered; when multiple GO IDs are reported, they represent all the categories contributing to the cluster, for which a single description is reported, corresponding to the category with the lowest *p*-value.

**Figure 3 ijms-24-03181-f003:**
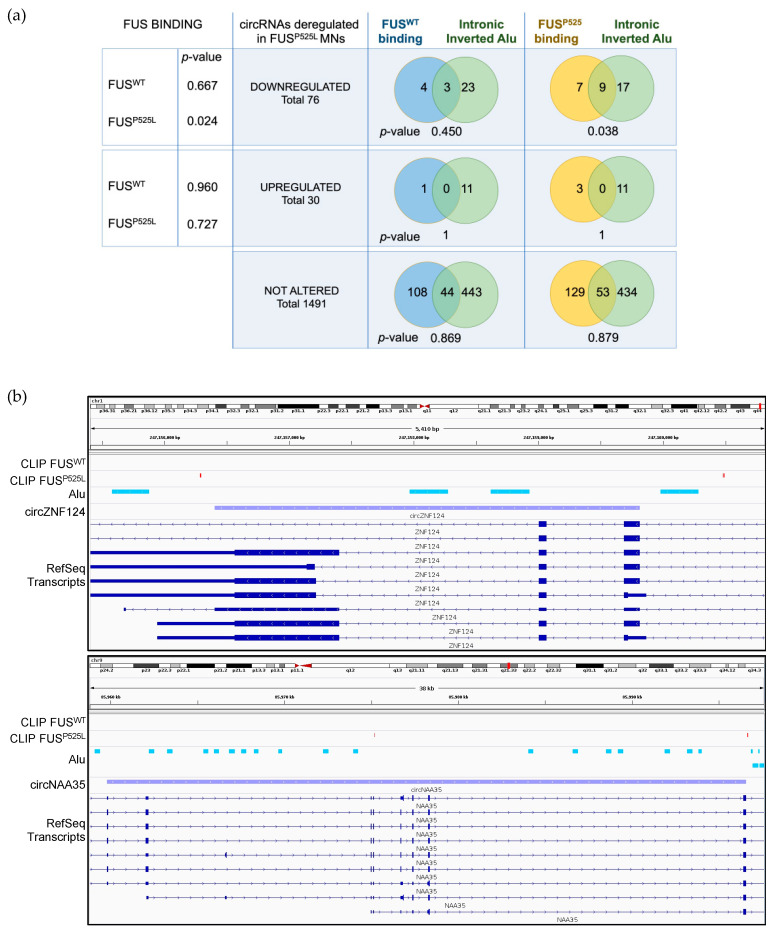
WT and mutant FUS binding in the intronic regions flanking motoneuronal circRNAs. (**a**) The left side of the table reports the one-sided Fisher’s exact test *p*-values describing the enrichment, with respect to unaffected circRNAs, of WT and mutant FUS binding sites (previously identified via PAR-CLIP; [44]) in downregulated (upper part) and upregulated (bottom part) circRNAs within 1 kb-long flanking intronic regions. The right side of the table reports the overlap between the circRNAs with WT or mutant FUS binding sites in the 1 kb flanking intronic regions and those hosting inverted Alu repeats in the same regions. CircRNAs were divided into downregulated, upregulated and unaffected upon FUS mutation. The deregulated circRNAs whose linear counterpart significantly varies in the same direction were excluded from this analysis, since it is less likely that FUS directly modulates their biogenesis. (**b**) IGV genome browser [60] tracks representing two circRNAs (circZNF124 and circNAA35) whose 1 kb-long flanking intronic regions contain both P525L FUS binding sites (red boxes) and inverted Alu repeats (light blue boxes).

**Figure 4 ijms-24-03181-f004:**
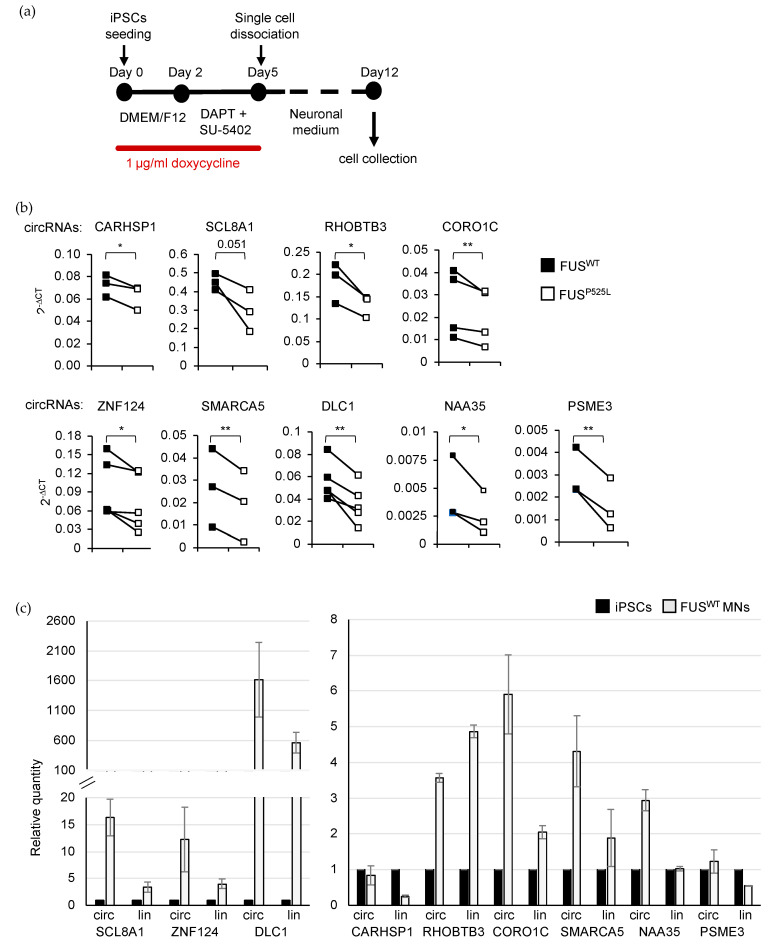
Validation of circRNA expression in FUS^WT^ and FUS^P525L^ MNs and alongside differentiation. (**a**) Schematic representation of the differentiation protocol used in this study. The medium and factors used along the timeline (day) of differentiation are indicated in the boxes while the red line depicts the duration of the treatment with doxycycline. The timing of MN collection for further analyses is also indicated. (**b**) Plots showing the levels of the indicated circRNAs measured by quantitative Real-Time PCR in FUS^WT^ and FUS^P525L^ MNs collected at day 12. CircRNA levels were quantified using ATP5O mRNA levels as reference and expressed as 2 ^-DCT. Values from all the biological replicates (*n* ≥ 3) are shown. *p*-values were calculated using paired one-tailed Student’s *t*-test (* *p* < 0.05 and ** *p* < 0.01). (**c**) Bar plot showing the levels of the indicated circRNAs measured by quantitative Real-Time PCR in iPSCs and FUS^WT^ MNs collected at day 12. CircRNA levels were quantified using ATP5O mRNA levels as reference and expressed as relative quantity with respect to iPSC samples set to a value of 1. Error bars represent s.d. of two independent experiments.

**Figure 5 ijms-24-03181-f005:**
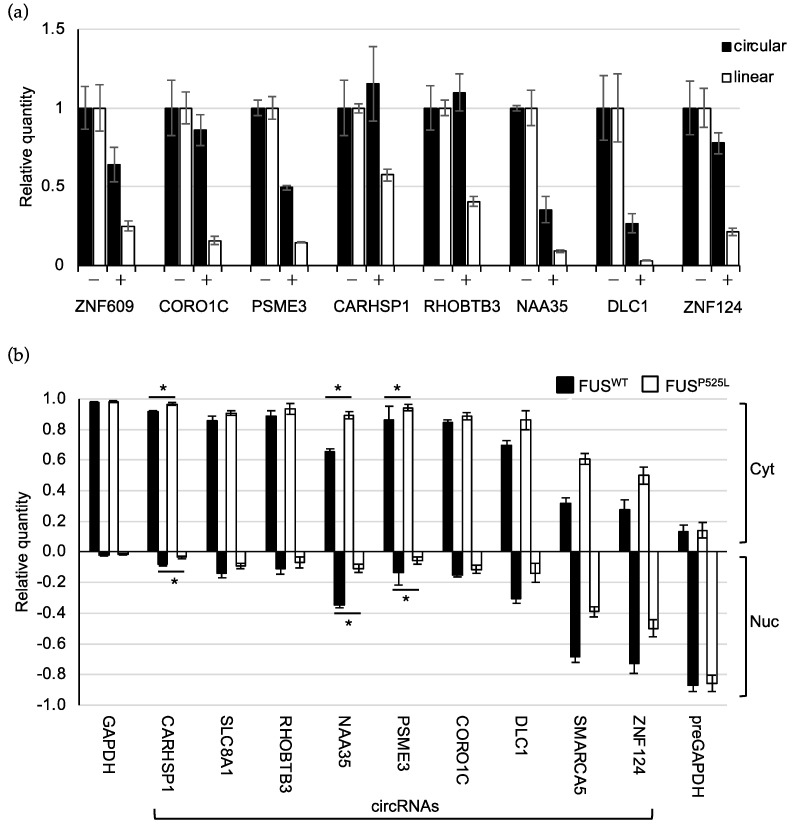
Circular molecules are resistant to RNaseR treatment and are mainly localized in the cytoplasm. (**a**) Bar plot showing the levels of the indicated circular and linear transcripts measured by quantitative Real-Time PCR upon the RNase R treatment (+) of total RNA from FUS^WT^ mature MNs. CircRNA levels were quantified using spike DNA levels as reference and expressed as relative quantity with respect to untreated samples (−) set to a value of 1. Circ-ZNF609 and its linear counterpart were used as the positive control for the activity of RNase R enzyme on circular and linear transcripts. Error bars represent s.d. of two independent experiments. (**b**) Bar plot showing the levels of the indicated circRNAs measured by quantitative Real-Time PCR in nuclear (Nuc) and cytoplasmic (Cyt) compartments of FUS^WT^ and FUS^P525L^ mature MNs. GAPDH and preGAPDH levels are used as controls for the quality of the fraction procedure. Error bars represent s.e.m. of at least three independent experiments. *p*-values were calculated using paired two-tailed Student’s *t*-test (* *p* < 0.05).

**Figure 6 ijms-24-03181-f006:**
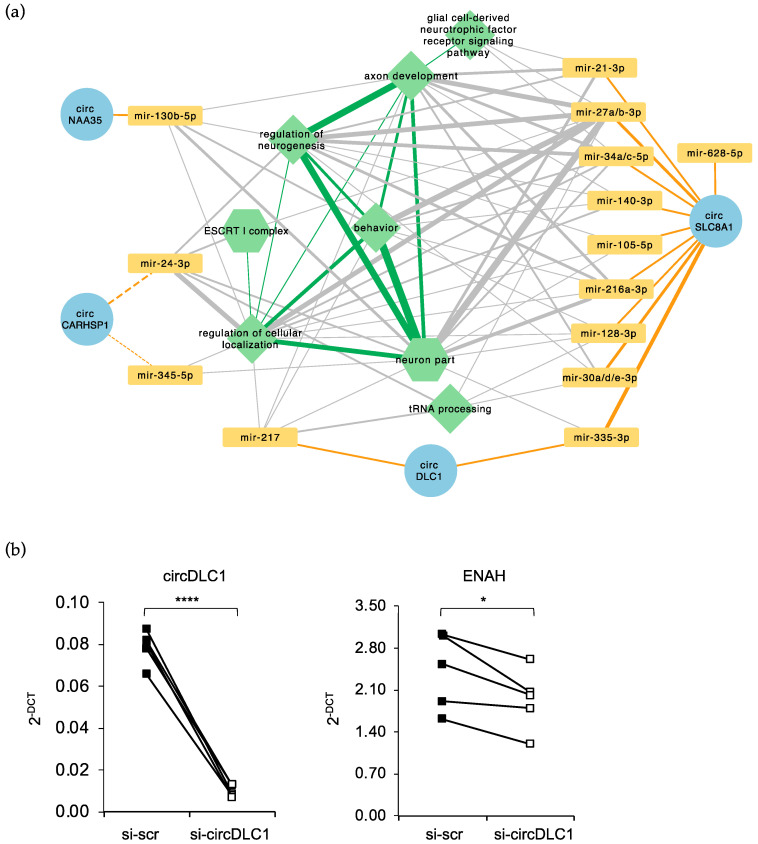
circRNA–miRNA–mRNA network inferred from high-throughput data. (**a**) The network reports the downregulated cytoplasmic circRNAs predicted to act as sponges for miRNAs expressed in MNs. Orange edges connect circRNAs to miRNAs, for which they have at least two predicted binding sites, or at least one binding region spanning the back-splicing junction (dashed lines), the edge width being proportional to the number of sites. Green nodes correspond to the GO Biological Process (hexagons) or Molecular Function (diamonds) terms enriched among all the known or predicted targets of sponged miRNAs (see Materials and Methods); only targets whose mRNA and/or protein are downregulated upon FUS mutation were used. Gray edges connect miRNAs to the functional categories to which at least one target belongs, the edge width being proportional to the number of targets. Green edges connect categories with at least one target gene in common, the width being proportional to the number of shared genes. (**b**) Plots showing the levels of the circDLC1 and ENAH mRNA measured by quantitative Real-Time PCR in MNs treated with si-circDLC1 and with si-scr as the control. RNA levels were quantified using ATP5O mRNA levels as the reference and expressed as 2 ^-ΔCT. Values from all the biological replicates (*n* = 5) are shown. *p*-values were calculated using paired two-tailed Student’s *t*-test (* *p* < 0.05 and **** *p* < 0.0001).

**Table 1 ijms-24-03181-t001:** List of circRNAs analyzed in this study.

GRCh38 Coordinates	Host Gene Name	Log2(FC)	*p*-Value
16:8858350-8859335_-	CARHSP1	−2.043424531	0.018454
5:95755396-95763620_+	RHOBTB3	−1.099679428	0.008357
12:108652272 108654410_-	CORO1C	−0.986079226	0.002601
1:247156406-247159813_-	ZNF124	−0.548012424	0.008117
4:143543509-143543972_+	SMARCA5	−0.368885114	0.023003
9:85959793-85996577_+	NAA35	−4.693947572	0.003701
17:42838731-42839380_+	PSME3	−4.31708731	0.008135
2:40428473-40430304_-	SLC8A1	−0.520421943	0.019316
8:13499049-13500196_-	DLC1	−0.409393486	0.053325

## Data Availability

This study used publicly available dataset. RNA-seq and PAR-CLIP raw data used in this study are available in the GEO and dbGaP databases with the accession numbers GSE94888, GSE203173, phs000747, and GSE118347. The raw LC–MS/MS data used in this study have been deposited to the ProteomeXchange Consortium with the dataset identifier PXD019596.

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
