# Peer review of "FUS Alters circRNA Metabolism in Human Motor Neurons Carrying the ALS-Linked P525L Mutation"

_ijms, 2023, doi:10.3390/ijms24043181_

Round 1
Reviewer 1 Report
In this study, Colantoni and colleagues performed a circular RNA (circRNA) analysis associated with FUS ALS mutation P525L in motor neurons. Initially, the authors performed a computational analysis of circRNAs using publicly available RNA-Seq data (NCBI GEO: GSE94888). The analysis revealed a fraction of circRNAs predominantly downregulated in the FUS P525L mutant compared to wt. Subsequently, the authors performed experimental verification of the altered expression of circRNAs in iPSC MNs by qPCR analysis from nuclear and cytoplasmic fractions. Findings suggested some circRNAs such as circNAA35 and circCARHSP1 significantly decreased in the cytoplasm and increased in the nucleus compared to the linear counterparts.
The authors are kindly asked to consider the following major and minor points to improve the manuscript.
Major points:
1- The workflow strategy of the circRNA computational analysis is extensive, but the major shortcoming is relying on only a single RNA-Seq data (NCBI GEO: GSE94888) which has a small sample size (n=16; triplicates of each condition). However, there is at least one other study (NCBI GEO: GSE172459; PMID: 35120624), which has a much larger dataset (n=334) and whose overall design largely intersects with the one used in the current study. It would have been a much more comprehensive and significant analysis if the authors would have considered this study too. It is strongly suggested to extend the current study, covering the above-mentioned RNA-Seq data and re-performing the circRNA analysis to reveal whether the final significant circRNA candidates are commonly shared in both studies. This will enable the authors to conclude more significantly on their relevance with P525L mutation and potential functions over ALS pathogenesis.
2- Did the authors test whether suppression of the expression of any or some of the final circRNAs reiterates the ALS cell phenotype in FUS wt cells?
3- As the authors presented in Table S3, the final novel circRNA candidates which might have miRNA sponging functions are circCARHSP1, circDLC1(not significant), and circNAA35. If their expression decreases, they will sponge fewer miRNAs, and it is expected that the expression of their target genes should be increased. Have authors happened to test any of them (via e.g. siRNA, shRNA, and overexpression) whether they sponge the predicted miRNAs and alter the expression of corresponding genes?
Minor points:
1- It is suggested that the text size and font should be the same and large enough for all the figures for better reading and representation. Also, the quality of the figures should be improved.
Author Response
Major points:
1- The workflow strategy of the circRNA computational analysis is extensive, but the major shortcoming is relying on only a single RNA-Seq data (NCBI GEO: GSE94888) which has a small sample size (n=16; triplicates of each condition). However, there is at least one other study (NCBI GEO: GSE172459; PMID: 35120624), which has a much larger dataset (n=334) and whose overall design largely intersects with the one used in the current study. It would have been a much more comprehensive and significant analysis if the authors would have considered this study too. It is strongly suggested to extend the current study, covering the above-mentioned RNA-Seq data and re-performing the circRNA analysis to reveal whether the final significant circRNA candidates are commonly shared in both studies. This will enable the authors to conclude more significantly on their relevance with P525L mutation and potential functions over ALS pathogenesis.
We thank the reviewer for this suggestion. Unfortunately, the RNA-Seq libraries from the suggested study (PMID: 35120624) were prepared using the KAPA Stranded mRNA-Seq Kit, thus allowing only the analysis of polyA+ RNAs. For this reason, they are not suitable for circRNA identification, since these molecules lack the polyA tail. Moreover, the study is focused on a human microglia model and, due to the tissue specificity of circRNAs, it is likely that the deregulated species would differ between the two different studies.
However, following the reviewer suggestion we searched for other total RNA-Seq data obtained from motor neurons carrying the P525L mutation, but, unfortunately, we could not find any public data suitable for circRNA identification.
We therefore decided to determine whether the deregulation of the identified circRNAs was a general feature of the ALS condition by analyzing two additional RNA-Seq data obtained from in vitro derived MNs carrying the H517Q FUS mutation ((NCBI GEO: GSE203173, PMID: 35750046) and from sporadic ALS post-mortem tissues (ventral horns of the lumbar spinal cord; dbGaP: phs000747, PMID: 28855684).
The results of these analyses are now described in paragraph 2.3 and shown in the new Figure S5. We observed a very mild overlap between the three datasets, with only two of the characterized circRNAs being in common between the FUSP525L and FUSH517Q conditions. Nevertheless, they show an opposite trend of deregulation. Moreover, it is worth noting that in FUSH517Q MNs and in sporadic ALS tissues a higher level of concordance was detected between the variation of circRNAs and of their linear counterparts, suggesting a major contribution of transcriptional regulation. Indeed, transcriptional defects in FUSH517Q MNs have been already described (https://doi.org/10.1016/j.stemcr.2022.05.019).
We can therefore conclude that P525L mutation causes a specific deregulation of a subset of circRNAs not observed in other ALS conditions and that this deregulation mainly occurs at the post-transcriptional level. It is important to underline that P525L mutation is linked to one of the most severe forms of ALS and it leads to an important delocalization of the protein in the cytoplasm, while H517Q, also located in the NLS of FUS, is a recessive mutation inducing a mild mislocalization phenotype (https://doi.org/10.1186/1750-1326-8-30).
We thank the reviewer for raising this issue, because we believe that the additional analyses strengthen the conclusion of the original manuscript.
2- Did the authors test whether suppression of the expression of any or some of the final circRNAs reiterates the ALS cell phenotype in FUS wt cells?
It is an interesting point to be addressed, indeed we are generating IPSCs KO for some of the selected circRNAs in order to identify altered pathways through omic analyses. However, this procedure takes time to be settled and we would like to include the data derived from KO cells in a new manuscript. We believe that this information is out of the aim of the present manuscript, that was meant to be a starting point for deeper investigation of circRNA function.
3- As the authors presented in Table S3, the final novel circRNA candidates which might have miRNA sponging functions are circCARHSP1, circDLC1(not significant), and circNAA35. If their expression decreases, they will sponge fewer miRNAs, and it is expected that the expression of their target genes should be increased. Have authors happened to test any of them (via e.g. siRNA, shRNA, and overexpression) whether they sponge the predicted miRNAs and alter the expression of corresponding genes?
As previously mentioned, investigating the function of selected circRNAs is the aim of our ongoing study. However, waiting for circRNA KO iPS clones, we tried to downregulate some circRNAs in MNs through siRNA treatment. We succeeded in obtaining a very strong downregulation of circDLC1, a circRNA predicted to sponge miR-217 and miR-335-3p.
We therefore selected some of predicted targets of these miRNAs: ADAM12 and EXOC7 targeted by miR-335-3p and HNRNPA3, ENAH and PDHA1 targeted by miR-217. We started to measure the alteration of their expression through Real-Time PCR. Among the miRNA targets only ENAH mRNA showed a significant downregulation when circDLC1 levels were reduced in FUSWT MNs. These results are shown on the new Figure 7b and discussed in the main text.
We are aware that miRNA-mediated translational regulation might not always be followed by reduction of mRNA levels, therefore, for the other mir-217 targets that do not vary in circDLC1 knock-down condition, further analyses of protein levels will be required. Moreover, regarding the mRNAs targeted by miR-335-3p, a cooperative mode of action of circDLC1 and circSLC8A1 has to be considered, therefore, the absence of only one of the two circRNAs might not be sufficient to determine an alteration of the target expression. These experiments have been performed by Sara D’Uva and she have been included in the author list.
Minor points:
1- It is suggested that the text size and font should be the same and large enough for all the figures for better reading and representation. Also, the quality of the figures should be improved.
We are sorry about the poor quality of the figures. We noticed that figure 1 is badly formatted, probably due to a conversion error. We adjusted the size and standardized the font of the figure text, and we improved the quality and resolution of the images.
Reviewer 2 Report
In the manuscript, the authors reported alterations of circRNAs levels in FUSP525L MNs and an impact to their nuclear/cytoplasmic partitioning.
The results are potentially interesting. However, I have some comments.
1- It lacks in the paper a validation of nuclear and cytoplasmic fractions (absence of contaminant nuclear proteins in the cytoplasmic fraction and vice versa). Therefore I suggest to show in supplementary material an analysis of the fractionation (probably already performed, by WB for example).
2- Student’s t test cannot be used to compare groups of n = 3 (e.g. in figure 4) and with a sample with no variance (the control). I suggest to use a non-parametric test (e.g.,Wilcoxon). If results are not significant, then it should be stated that non statistically significant trends were identified (trends are still valid as long as used to guide confirmatory experiments).
Author Response
In the manuscript, the authors reported alterations of circRNAs levels in FUS P525L MNs and an impact to their nuclear/cytoplasmic partitioning.
The results are potentially interesting. However, I have some comments.
1- It lacks in the paper a validation of nuclear and cytoplasmic fractions (absence of contaminant nuclear proteins in the cytoplasmic fraction and vice versa). Therefore, I suggest to show in supplementary material an analysis of the fractionation (probably already performed, by WB for example).
We apologize for not clearly mentioning this in the main text. Indeed, since we extracted RNA, we used GAPDH mRNA and pre-mRNA to validate the efficiency of the nuclear/cytoplasmic fractionation (see legend of Figure 5). This is now better explained in the revised manuscript.
2- Student’s t test cannot be used to compare groups of n = 3 (e.g. in figure 4) and with a sample with no variance (the control). I suggest to use a non-parametric test (e.g.,Wilcoxon). If results are not significant, then it should be stated that non statistically significant trends were identified (trends are still valid as long as used to guide confirmatory experiments).
We thank the reviewer for highlighting this point. Actually, the control samples (2^-DDCT values) do have a variance and we applied error propagation when we computed the means. However, since our sample size is too small to apply Wilcoxon test that, indeed, would be not significant anyway with n=3 or 4, following the reviewer suggestion we decided to present the data as 2^-DCT showing the results of each single experiment; moreover, we applied one tail t-test, as downregulation is expected, using a paired test. This latter would strengthen the statistical power of the test. The new Figure 4b showed that, except circSLC8A1 which has a p-value close to significance, all the other circRNAs reach significance. This is now stated in the revised version of the manuscript.
Round 2
Reviewer 1 Report
I thank the authors for considering my questions and comments to improve the manuscript. The revised version of the manuscript is suggested for publication.